# Validation of a Temperature-Feedback Controlled Automated Magnetic Hyperthermia Therapy Device

**DOI:** 10.3390/cancers15020327

**Published:** 2023-01-04

**Authors:** Anirudh Sharma, Avesh Jangam, Julian Low Yung Shen, Aiman Ahmad, Nageshwar Arepally, Benjamin Rodriguez, Joseph Borrello, Alexandros Bouras, Lawrence Kleinberg, Kai Ding, Constantinos Hadjipanayis, Dara L. Kraitchman, Robert Ivkov, Anilchandra Attaluri

**Affiliations:** 1Department of Radiation Oncology and Molecular Radiation Sciences, The Johns Hopkins University School of Medicine, Baltimore, MD 21231, USA; 2Department of Mechanical Engineering, School of Science, Engineering, and Technology, The Pennsylvania State University—Harrisburg, Harrisburg, PA 17057, USA; 3Sinai BioDesign, Mount Sinai Hospital, New York, NY 10029, USA; 4Department of Neurosurgery, Icahn School of Medicine at Mount Sinai, New York, NY 10029, USA; 5Department of Neurological Surgery, University of Pittsburgh School of Medicine, Pittsburgh, PA 15213, USA; 6Department of Oncology, Johns Hopkins University School of Medicine, Baltimore, MD 21287, USA; 7Department of Radiology and Radiological Science, Johns Hopkins University School of Medicine, Baltimore, MD 21205, USA; 8Department of Mechanical Engineering, Whiting School of Engineering, Johns Hopkins University, Baltimore, MD 21218, USA; 9Department of Materials Science and Engineering, Whiting School of Engineering, Johns Hopkins University, Baltimore, MD 21218, USA

**Keywords:** magnetic hyperthermia, magnetic nanoparticles, cancer nanomedicine, iron oxide nanoparticles, automated temperature control, proportional-integral-derivative controller

## Abstract

**Simple Summary:**

Magnetic hyperthermia therapy (MHT) is a promising nanotechnology-based treatment for cancer. Its widespread adoption is hampered by the imprecise control of tumor temperature during treatment and by a lack of automatic shutdown to avoid adverse events. Here, we verify the functionality and validate performance of an automated temperature controller, and the functionality of automated safety shutdowns. Experiments were performed ex vivo in liver tissue and in vivo in the brain of a healthy live research subject to demonstrate the potential for superior energy control with temperature stability using user-defined inputs, and real-time temperature monitoring for feedback. Performance of this device was validated against design criteria using FDA guidelines. It will be used to treat canine glioblastoma tumors in a future study.

**Abstract:**

We present in vivo validation of an automated magnetic hyperthermia therapy (MHT) device that uses real-time temperature input measured at the target to control tissue heating. MHT is a thermal therapy that uses heat generated by magnetic materials exposed to an alternating magnetic field. For temperature monitoring, we integrated a commercial fiber optic temperature probe containing four gallium arsenide (GaAs) temperature sensors. The controller device used temperature from the sensors as input to manage power to the magnetic field applicator. We developed a robust, multi-objective, proportional-integral-derivative (PID) algorithm to control the target thermal dose by modulating power delivered to the magnetic field applicator. The magnetic field applicator was a 20 cm diameter Maxwell-type induction coil powered by a 120 kW induction heating power supply operating at 160 kHz. Finite element (FE) simulations were performed to determine values of the PID gain factors prior to verification and validation trials. Ex vivo verification and validation were conducted in gel phantoms and sectioned bovine liver, respectively. In vivo validation of the controller was achieved in a canine research subject following infusion of magnetic nanoparticles (MNPs) into the brain. In all cases, performance matched controller design criteria, while also achieving a thermal dose measured as cumulative equivalent minutes at 43 °C (CEM43) 60 ± 5 min within 30 min.

## 1. Introduction

Magnetic hyperthermia therapy (MHT) is a powerful nanotechnology-based treatment that can enhance the effectiveness of standard-of-care therapies, such as radiation therapy (RT) and chemotherapy (CT) [1]. MHT involves intratumoral delivery of a magnetic colloid, i.e., magnetic nanoparticles (MNPs) suspended in biocompatible fluid, followed by application of an alternating magnetic field (AMF) to generate local hyperthermia (HT) [2,3]. One potential advantage of MHT over other modalities for brain malignancies, e.g., glioblastoma, is the potential to apply multiple fractions of HT over time, with relative reproducibility [4]. HT within the temperature range of 41 °C to 46 °C generates cell death by necrosis and/or apoptosis [5,6,7] and can potentiate the tumoricidal effects of RT before or after treatment depending upon the thermal dose (time at temperature) [8,9,10,11,12]. It has been hypothesized that HT may transform relative biological effectiveness of low linear energy transfer (LET) RT to be equivalent to the effectiveness observed with high LET RT because it often improves tumor oxygenation effectively reducing oxygen concentration-dependent effects of ionizing radiation [13,14]. Optimal HT improves local tumor response and overall survival in patients when combined with either RT, CT, or their combination (CRT) [15,16,17].

In general, thermal dose is a dynamic process in tumors that are often heterogeneous, but methods have been developed to convert the dynamic time and temperatures achieved into an equivalent isoeffect dose defined as cumulative equivalent minutes (CEM) of exposure referenced to the thermal breakpoint temperature 43 °C. A common clinical objective is to achieve a minimum therapeutic thermal dose, CEM43 of about 30 min in 90% of the tissue volume, or CEM43T90 30 min. This is similar to the concept of equivalent dose of 2 Gy (EQD2) used in RT when dose per fraction and number of fractions vary in prescription. In the application of HT to patients, a means to validate a prescribed treatment plan to achieve the target thermal dose of CEM43T90 is needed [18,19,20]. Though rarely indicated as a single-agent therapy, achieving a thermal dose CEM43T90 >30 min with HT alone generally correlates with favorable treatment outcomes for many solid tumors independent of HT modality [19].

Conversely, quantitative correlations of thermal dose with clinical response following MHT are less established, in part because validated dose control devices are uncommon. While it is an advantage of MHT that MNPs produce heat locally within tissue this, however, can be a disadvantage when MNP distribution in tissue is heterogeneous, unknown, or if the MNPs leak out of the target. Furthermore, imprecise temperature and thermal dose control within the tumor and at tumor margins are often an unavoidable consequence arising from heterogeneous MNP distributions and leakage from tissue. Constant-power MHT, keeping AMF power fixed for the duration of treatment, though common in preclinical studies with small animal models, often fails to achieve the CEM43T90 objective in the target and can produce ablated healthy tissues near the tumor boundary [21,22,23,24,25]. Thus, while MNPs produce heat locally within tissue, nanoparticle heating and off-target (Joule) heating by induced eddy (Foucault) currents [26,27] must be considered. 

Small animal and computational models have demonstrated that amplitude and power modulation can compensate for heterogeneous MNP distribution to improve MHT [21,26,28]. Various strategies have been proposed with computational models to achieve better control of energy deposition. Ahmed et al. studied closed-loop temperature control using rotating AMFs to initiate formation of MNP chains that could be magnetically guided to concentrate in the region of interest [29]. Singh et al. have attempted to model MHT with dynamic MNP distribution using a modified tissue porosity model [24]. Recognizing that MNP distribution is critical to accurately modeling the temperature distribution, Hedayatnasab et al. applied a particle swarm optimization to an artificial neural network model to explore optimal nanoparticle heating properties, MNP tissue concentration, AMF conditions and heating times [30]. Using 2D and 3D computer phantoms, Kandala et al. predicted improved thermal dose distribution in tumor with proportional-integral-derivative (PID)-controlled amplitude modulation with feedback provided by temperature measured at the tumor-tissue boundary [21,28]. Though PID-based temperature control is common in many industrial and consumer products, it is less so for medical applications of MHT [29,31,32,33,34].

To implement PID-based treatment in clinical-scale MHT systems, appropriate hardware and software integration, and device verification and validation are required. Moreover, while in silico temperature feedback-controlled experiments are invaluable to optimize experimental parameters, such as probe placement, optimal AMF for CEM43T90, and AMF safety limits [21,35,36,37], computational modeling and simulations-based (CM&S) treatment planning assume a priori that the temperature controller operates ideally and meets performance requirements. In practice, these device-related assumptions must be verified and validated to confirm performance, safety, and reliability [38].

An ideal temperature feedback controller for MHT achieves target temperature rapidly and maintains temperature by adjusting power to compensate for changes, e.g., blood perfusion, during treatment to achieve a prescribed thermal dose. Additionally, the controller limits or disconnects power if off-target heating threatens healthy or sensitive tissues. While conventional ON-OFF switch-based temperature controllers are used in preclinical models with laser ablation [32,33], PID controllers provide superior control over the transient (dynamic) and steady-state temperature-time evolution. In addition, a controller can filter high-frequency sensor noise. Favorable dynamic characteristics include (i) short time to reach the set point temperature (rise time, *t_r_*); (ii) short settling time (*t_ss_*), which is the time taken for the controller transients to decay; and, (iii) small overshoot (*M_p_*), which is the temperature that exceeds the set point temperature. Favorable steady-state characteristics include (iv) small steady-state error and, (v) robust temperature control against variations in temperature within the target and surrounding tissue. These characteristics can be controlled by tuning the proportional, integral, and derivative gains (*K_p_*, *K_i_*, *K_d_*) of the PID controller. Proportional control allows power control proportional to the difference between measured and set point temperature, as found in sound amplifiers. Integral control minimizes the steady state error between measured and set point temperatures. Derivative control improves transient characteristics, offering faster rise times and low overshoot [39]. When used together, PID temperature control offers precise temperature control balancing transient temperature excursions with steady-state control during heating. Such precision in temperature control is especially important for MHT where quantitative correlations between prescriptive thermal dose CEM43 (time-at-temperature) and treatment outcomes are necessary.

MHT was approved in 2010 by the European Medicines Agency to treat recurrent glioblastoma (GBM) with fractionated RT. GBM remains a largely fatal brain cancer that is resistant to standard therapies. Standard-of-care includes surgical resection combined with adjuvant fractionated external beam RT and concomitant temozolomide (TMZ) CT [40,41,42,43,44,45,46]. Nonetheless, poor outcomes persist because residual and viable tumor cells arising from blood–brain barrier (BBB) limitations to drug delivery, tumor cell resistance to CRT, and tumor molecular heterogeneity spur recurrence into the surrounding brain [47,48,49,50]. MHT may enhance GBM treatment since it sensitizes tumor cells to RT and CT. Understanding of mechanisms explaining tumor cell sensitizing HT + RT or HT + CT combinations has improved. These include DNA damage repair inhibition, increased blood flow and transient opening of the BBB to enhance treatment of invasive tumor cells in the surrounding brain [51,52,53,54]. MHT sensitizes therapy-resistant glioma stem cells (GSCs) known to mediate tumor recurrence after RT [54]. MHT is being explored for other solid tumors such as prostate [55,56,57], pancreas [27,58,59], bone [60] and liver [25,61,62].

While MHT is clinically validated for treating GBM, and substantial preclinical evidence demonstrates its versatility and potential benefits, its clinical adoption and translation to other indications is inhibited by challenges to achieve reliable control of thermal dose with prescriptive therapy similar to RT. Typically, energy control is performed manually with single-point thermometry, or it is uncontrolled with constant power. Herein, we document the design and validation of a PID-based temperature feedback control system for controlling heating in tissues in a scaled-up, 120 kW AMF system coupled to a custom-designed 20 cm diameter RF coil with uniform AMF over a 3L volume of interest (VOI) [63]. We designed the controller to reach user-selected target temperature in a tissue within 60 s with minimal overshoot (<5%) and to maintain a user-selected target temperature for 15–30 min with a steady-state error <1%. Initial PID gain values derived from simulations were adjusted using experimental values. Safety limits were defined in controller operation to limit overtreatment by runaway MNP heat outside the target volume and to limit off-target heating from induced eddy currents. In vivo validation was performed in a canine research subject, where MNPs injected into the brain were used as the primary heat source to validate temperature control during brain tissue heating. Design input and output definitions, design verification, review and validation were performed using the FDA Device Control Guidance for Medical Device Manufacturers [38].

## 2. Methods and Materials

### 2.1. Power Supply and Coil

A 20 cm diameter modified Maxwell coil (AMF Life Systems, Auburn Hills, MI, USA), connected to a 120 kW induction heating power supply (PPECO, Watsonville, CA, USA) previously described was used for generating the AMF (Figure 1) [63]. The power supply, matching networks, and coils were cooled with a closed-loop circulating water system maintained at 22 °C to 28 °C (Dry Cooler Systems, Inc., Auburn Hills, MI, USA). A commercially available two-dimensional magnetic-field probe (AMF Life Systems, Auburn Hills, MI, USA) was used to calibrate field amplitude, *H*, vs. voltage, *V*, and field uniformity inside the coil [63,64]. The magnetic field was uniform (<10% variation) in a cylindrical volume with dimensions of 10 cm length and 16 cm diameter, for AMF amplitudes up to 35 kA/m at 155 ± 10 kHz.

### 2.2. Optical Fiber Thermometry and Multi-Sensor Probe

Temperature measurements were performed using fiber optic temperature probes comprising four gallium arsenide (GaAs) temperature sensors spaced 2 cm apart along the length of a 0.8 mm diameter optical fiber (FISO Technologies, Ltd., Quebec, QC, Canada). The manufacturer calibrated the probe. It provided readings with accuracy ±0.3 °C, resolution 0.1 °C, and response time <100 ms. A FISO EVO-RM-8 Rack-Mount (FISO Technologies, Ltd.) signal generator, along with four FISO SPC-HR (FISO Technologies, Ltd.) modules, with analog output (AO) ports were used to read temperature signals from the fiber optic probes and transfer signals from the analog output ports to the controller.

### 2.3. Controller Design

Our objective in this effort was to validate a temperature control device to automate MHT of spontaneous canine gliomas. Fully automated treatment to a prescribed plan requires real-time temperature feedback. Additional temperature monitoring at sites distant from target enabled with override function is also needed to minimize overtreatment outside the target and to reduce risks of excessive power deposition from induced eddy current heating. A typical setup for controlling temperature during tissue heating or MHT includes a heat source (e.g., MNPs or metallic seed implant), temperature probe sensor(s) for feedback control, at least one “safety” temperature probe sensor for monitoring the core body temperature during the MHT treatment, and a second safety temperature probe to monitor non-specific heating from induced eddy currents at the tissue periphery or surface of the region exposed to AMF [57,65]. MHT treatment and controller performance criteria we selected included achieve and maintain temperature at a target set point within the hyperthermia range (43–45 °C) for clinically relevant time 15–30 min to attain CEM43 of 60 ± 5 min at target probe location; attain *T_ref_* quickly (rise time, *t_r_* < 60 s); minimize temperature overshoot (*M_p_* < 5%); and, achieve a settling time (*t_ss_* ±0.5 °C < 5 min).

#### 2.3.1. Safety Controls

To ensure safety, a maximum temperature at the treatment region (e.g., 50 °C) that limits power to prevent runaway heating at the feedback sensor location, and a safe temperature threshold at a distant location, such as core or tissue surface temperature for additional safety monitoring are required. We integrated independent power-limiting (user-controllable) safety thresholds into the design to ensure alignment with FDA CDRH guidelines on device safety [38]. Feedback and safety temperature sensors were designed to work independently to limit power. To meet safety design requirements, a LabVIEW (National Instruments, Austin, TX, USA) program was written to include a user-defined safety temperature, which powers down the device if exceeded. To limit non-specific tissue heating due to eddy currents and to prevent core body temperature increases during treatment, the user specifies power limiting safety temperature thresholds. The safety temperature probe was connected to a separate recording system (FISO TMI4 FISO, Quebec, QC, Canada) to further enhance safety control.

#### 2.3.2. Hardware and Software

For reading temperature (from FISO EVO analog output) and controlling power, a NI CompactRIO (NI cRIO 9042 1.6 GHz quad-core) with the multifunction I/O module (NI9381, with 8 analog inputs and 8 analog outputs) and compatible LabVIEW software interface (National Instruments, Austin, TX, USA) were used. Digital signal processing, error estimation, and conversion of the error signal to the appropriate control signal (0–5 V) for the 120 kW AMF system were performed in LabVIEW. Temperature, power, and CEM43 data were exported from the cRIO controller in TDMS format.

### 2.4. Calculation and Verification of PID Gains ($K_{p},K_{i},K_{d})$ for Temperature Feedback Control

Initial PID gains (*K_p_*, *K_i_*, *K_d_*) for closed loop feedback control were derived from computer simulations of an agarose gel + Cu (wire) and these were verified experimentally. As no standard reference materials (SRMs) are available for MNPs, Cu wire SRMs can be used for verification of RF-induced heating in vitro [64]. The test system was adapted from Attaluri et al. [64]. Briefly, 1 mL of 1% agarose in an Eppendorf tube and a NIST traceable Cu standard wire segment (3 N purity; ESPI Metals, Ashland, OR, USA) weighing 0.104 g, placed at the center of the gel surface were used and simulated (Figure 2A,B and Appendix A). The fiber optic temperature probe was placed at a fixed distance (1.3 mm) from the Cu wire. Additional details on calculations and modeling are provided in a separate publication [21,66]. Controller inputs were calculated for using inputs (Appendix A) to achieve set point temperature of 25 °C, rise time <1 min, with initial overshoot <5%, and duration 15–30 min to reflect treatment times. In vitro validation experiments for temperature controller response were conducted three times (N = 3).

These performance criteria can be equivalently achieved for MNPs if the total power deposited by the MNPs is the same per unit volume, under the same AMF amplitude and frequency, i.e.,
(1)$$P=SLP\times c_{Fe}=SAR_{Cu}\times \rho_{Cu}$$
where $c_{Fe}$ are the concentration of Fe from MNPs and $\rho_{Cu}$ is density of Cu wire. *SAR_Cu_* is the specific absorption rate of Cu wire of a fixed mass and *P* is the power density in W/m^3^.

### 2.5. Ex Vivo Validation of Controller Performance for MHT in Bovine Liver Tissue

An ex vivo bovine liver lobe section (lobe) was maintained at 23 ± 1 °C to experimentally validate performance of the feedback controller. For tests, one or three sections of the standard reference material Cu wire were embedded as heat sources (Appendix A). To simulate a localized MNP distribution as from direct intratumor injection, a single Cu wire was embedded in the center of the largest surface with temperature probe 1.5 ± 0.1 mm away (Figure 3A). To simulate distributed MNP localization, three Cu wire sections were embedded in a triangular configuration into the largest volume of the liver with temperature probe placed in the center (Figure 3B). For all experiments, a second temperature probe representing the safety probe was placed at the liver periphery. Controller PID gain values ($K_{p}=0.5,K_{i\;}=1\times 10^{-6},K_{d}=2.5)$ were tested with temperature set points, *T_ref_* = 44 °C, 44.5 °C and 45 °C for 30, 22.5 and 15 min, respectively, in separate experiments for single or distributed heat source configurations to achieve a target CEM43 = 60 min (±5 min). Experiments were conducted for two different scenarios with three experiments escalating temperature for each, single heat source (N = 3) and distributed heat sources (N = 3). Within each scenario (single or distributed heat source), each experiment tested the controller performance to achieve one of a target temperature set point, *T_ref_* = 44 °C, 44.5 °C or 45 °C, respectively. Safety probes settings were 40 °C at the liver periphery and 50 °C at a location near the control sensor. Temperatures were recorded at 1 s intervals and exported in TDMS format.

### 2.6. In Vivo Validation of Controller Performance for MHT in a Canine Research Subject

All studies were approved by the Johns Hopkins Institutional Animal Care and Use Committee (IACUC). For in vivo validation, a healthy, adult male beagle (~10 kg) with a custom catheter for MNP delivery in left frontal lobe of the brain was studied. All studies were performed under general anesthesia. Briefly, an intravenous catheter was placed and the dog was sedated with fentanyl, induced with midazolam and propofol, intubated, and maintained with sevoflurane anesthesia with mechanical ventilation. A cone-beam computed tomography (CBCT, Artis Zee, Siemens, Malvern, PA, USA) was acquired after MNP infusion prior to MHT and used for heat-transfer simulations. Heat-transfer simulations on CT image-segmented brain were performed using COMSOL Multiphysics using methods previously published [21,27,35].

Prior to MHT, fiber optic probes were placed into the infusion port for the MNPs or a catheter port ~5 mm away from the infusion port, as well as subcutaneous tissue in the head and the rectum (Figure 4A). Probes in the catheter ports were used for MHT feedback control. Subcutaneous and rectal probes were used as additional safety probes to limit tissue surface and core temperature heating to 45 °C and 40 °C, respectively.

MHT was performed in two separate heating sessions on consecutive days. A total of three, short duration (<30 s) AMF pulse tests and three, longer temperature-control experiments (N = 3) were performed. For the latter cases, controller performance in achieving target temperatures of 39 °C and 45 °C was tested. The head of the dog was positioned in the center of the coil where the magnetic field is spatially uniform. Short duration pulses (<30 s) with varied (peak) amplitudes of 4, 8 and 12 kA/m were used to estimate initial PID gain values and to validate safety of our setup before commencing longer exposures. This was followed by conducting two separate 10 min PID-based temperature control experiments to achieve setpoints of 39 °C and 45 °C, respectively, at the MHT probe location. The lower setpoint of 39 °C was used initially as a safety precaution to assess response to a 10 min long AMF exposure, by monitoring the core (rectal) temperature, vital signs, and subcutaneous temperature for eddy current heating (Figure 4B and Appendix A). Next, the controller response to a higher setpoint of 45 °C was tested, and temperatures at all locations were recorded. An upper limit of 8 kA/m peak AMF amplitude was used. Based on the temperature vs. time data obtained from these 10 min experiments, PID gains of the controller and upper limit of peak AMF were adjusted for the next validation experiment. Specifically, proportional and integral gains were adjusted to achieve faster rise time and low steady-state error. Final gains of (2, 1 × 10^−4^, 2.5) were used. Upper limit of peak AMF was adjusted to 10 kA/m at 160 kHz. With these settings, a validation trial with setpoint 45 °C was conducted for 15 min. Complete blood count and chemistry were acquired prior to a second MHT study two days later. The dog was humanely euthanized after the second study, and a full necropsy was performed.

## 3. Results

### 3.1. In Vitro Verification

Simulations of the Cu-wire/gel phantom system yielded initial gains values (0.2, 0.001, 3) that produced oscillations (Figure 2D). Additional calculations yielded optimized values (0.23, 0.0001, 2.84) that resolved these oscillations (Figure 2F), and agreed with the controller design criteria (methods Section 2.3). The results from the trials were initial overshoot < 5%, $t_{r}$ = 8.35 s, and $t_{ss}$ = 39.7, all within design specifications.

### 3.2. Controller MHT Performance Validation in Ex Vivo Bovine Liver Tissue

In both geometric configurations (Figure 3A,B), rise time, settling time and overshoot met the original controller design criteria (Figure 3C), demonstrating robust control despite placement variations between heat source and temperature probe sensor. A target CEM43 of 60 ± 5 min was achieved. We observed longer rise times and settling times for distributed heat sources compared to single heat sources because the gains required optimization for three heat sources compared to a single source, forcing a compromise between stability and settling time.

### 3.3. Performance of Safety Controls

When temperature at the sensor location exceeded user-defined safety limits (e.g., with poorly optimized inputs or thermal runaway situations), the AMF system powered down and resumed when temperature fell below the safety limit, validating controller prioritizing safety (Appendix A).

### 3.4. In Vivo Validation of Controller for MHT in a Canine Research Subject

Cone beam CT images of the canine brain following infusion of the MNPs showed the location of the infusion catheter, and it was inferred that the MNPs were localized near the catheter tip and surrounding the catheter walls in the brain (Figure 4B). Finite element electromagnetic and heat transfer simulations were performed using the segmented CT image. Simulations with MNP heat sources predicted expected maximum temperatures near the catheter tip and walls, with rapid declines away from the heat source (Figure 4C). The control temperature sensor was placed at the interface of the catheter tip and tissue to measure tissue heating (Figure 4A). AMF pulses having duration 30 s, showed a non-linear increase in rate of temperature increase at the MNP sensor location (Appendix A). In trials lasting 10 min, a 39 °C setpoint was achieved within 200 s (Figure 4D). Temperature fluctuations (<0.4 °C), however, were observed at steady state. Rectal temperature remained stable at about 37 °C throughout the test (Figure 4D, orange line). Increasing temperature at the subcutaneous location, an indicator of tissue heating from eddy currents, did not exceed 38 °C (Appendix A).

In our initial 10 min exposure with setpoint of 45 °C, steady state at 45 °C in the brain tissue was not achieved initially (Figure 4E). The temperature at the MHT sensor location increased to about 44 °C and fluctuated between about 43 °C to about 44 °C. We attributed the fluctuating response to unoptimized PID gains, which is expected with poorly tuned controllers. Core body temperature (rectal) was stable at about 37 °C (Figure 4E, orange line). Temperature at the subcutaneous location increased from 36 °C to 38 °C during the 10 min treatment (Appendix A). Fluctuations in temperature vs. time at the subcutaneous location lagged response to power modulation because the PID controller was untuned. These 10 min AMF test results indicated that although PID gains $\left(K_{p},\;\;K_{i\;},K_{d}\right)$ were potentially sufficient for a setpoint of 39 °C, the gains required further tuning to arrive at a 45 °C hyperthermic setpoint. Specifically, we increased proportional gain *K_p_* and integral gain, *K_i_* to reduce the rise time and steady-state error to 45 °C.

The in vivo experiment was repeated on another day using a target setpoint of 45 °C after adjusting PID gain values $\left(K_{p},\;K_{i\;},K_{d}\right)$ to (2, 1 × 10^−4^, 2.5) and was maintained for 15 min (Figure 4F). In this case, the setpoint was achieved in about 120 s, with steady state error Δ*T* < 0.1 °C throughout the treatment. No overshoot or fluctuations were observed beyond 120 s. Thus, the 120 s equilibration time fell within the 5 min settling time criteria. More important, a CEM43 56 min was achieved, rectal temperature remained at 38–39 °C, and subcutaneous temperature at the head was 38–43 °C (Appendix A). Although this latter temperature sensor was placed at a subcutaneous location to monitor heating from eddy currents, some uncertainty of its placement, relative to the MNPs, was evident. Nevertheless, heating never exceeded the upper cutoff limit of 45 °C for safety. All vital signs remained normal, and the subject recovered completely between trials 1 and 2.

## 4. Discussion

Controlled energy deposition into the target remains a challenge for clinical applications of HT. Given MNPs are the heat source(s) for MHT, challenges to control energy delivery arise from their variable distributions within target and surrounding tissues. Interactions between electromagnetic (EM) fields and tissues generate off-target heating via induced eddy currents, further challenging energy control. Method of MNP delivery can significantly influence their tissue distribution. Most clinical applications rely on stereotactic injection of MNPs intratumorally which can result in MNP leakback, leak out, and inadequate distribution within the tumor [67,68]. Convection enhanced delivery (CED) has been tested in spontaneous canine gliomas and has shown promise. With this approach, direct intratumoral MNP delivery relies on a pressure dependent gradient [68,69,70,71]. However, variable MNP distribution following CED remains unavoidable as interactions between MNP composition, fluid dynamics, and individual tumor variations dominate MNP distribution in tissue. The consequence for MHT is variable or unpredictable thermal dose control to target and surrounding normal tissues.

Here, we document successful validation of a PID-based, multi-sensor temperature feedback controller and its integration with a 20 cm diameter RF coil intended for MHT treatments of gliomas in large animals. We verified controller performance in vitro in agarose gel + Cu wire model (Figure 2), and validated its performance by achieving CEM43 of 60 ± 5 min with steady temperature within a clinically relevant time (15–30 min) ex vivo (Figure 3) and (15 min) in vivo MNPs as heat sources in a live research canine (Figure 4). Differences between simulation and experiment were compensated by fine tuning the PID gain parameters experimentally.

For our future application of the temperature controller for MHT of spontaneously occurring GBM in canine patients, our study represents an especially important step because temperature control was validated in a model where heat was generated from heterogeneously distributed MNPs, against the backdrop of temperature-dependent blood perfusion, metabolic heat generation and anesthesia. The robustness of our controller against these variables demonstrates that the temperature controller can potentially be used to treat other solid tumors (e.g., hepatocellular carcinoma (HCC), pancreatic and prostate cancers), and it can be used in other biomedical heating applications which require temperature control.

We note that time- and temperature-dependent changes in heat transfer processes in vivo place a demand on the user for optimal tuning of the controller PID gains. In contrast, the dose of ionizing radiation absorbed by tissue during RT can be precisely calculated as absorbed dose from high energy X-rays used in therapy is closely associated with the absorption of kilovoltage radiation, and can be non-invasively measured in three dimensions with standard CT scanning. By contrast, there is no corresponding method to accurately predict energy absorbed in tissues for HT. Hence, a method of feedback control for MHT is necessary to ensure spatially localized and homogenous control of energy deposition that aligns with the treatment target, confirmed by measuring temperature in real time within tumor and margins. Further, energy absorption from heating dynamically changes during an individual treatment based on heat-related changes in tissue properties, perfusion, and eddy currents.

Computational modeling and simulations are integral to estimate initial PID gain values to guide initial heating of the target. Accuracy of estimated values can be improved by incorporating imaging data that provides MNP localization and concentration, e.g., magnetic particle imaging (MPI), co-registered with anatomical data, as from magnetic resonance imaging or X-ray CT. Even so, real-time adaptation of PID gains will be necessary to compensate for uncertainties in heat generation, mass and energy transfer, and heat loss and/or tissue degradation during treatment arising from stochastic damage events, patient-to-patient variability, and practical constraints on power supply dynamics.

In this in vivo validation test, we adapted the initial PID gains values to new values using in vivo pulsed and short-duration AMF trials. Specifically, we measured the rate of increase in temperature near the MNPs in the canine brain during 30 s AMF pulses (Appendix A), and steady state temperatures during 10 min PID trials at 39 °C and 45 °C setpoints (Figure 4B,C). From the results, we concluded that the calculated gains needed further tuning to decrease rise time (increasing *K_p_*) and reduce steady state error (increasing *K_i_*). While we obtained temperature readouts from multi-sensor temperature arrays, our controller was designed to use input from only one sensor for temperature control. Real time temperature feedback from multiple temperature sensors can potentially be used to develop adaptive control algorithms to improve MHT. Future efforts will focus on integrating adaptive PID algorithms with multi-input, multi-output (MIMO) feedback control.

Another area of development is adaptive magnetic hyperthermia (AMHT) achieved by integrating the adaptive PID algorithms with MIMO feedback control and imaging. Similar to adaptive radiation therapy (ART) which is widely practiced, our AMHT paradigm will be categorized into offline and online processes. In an offline process, before the patient is sent to treatment, anatomical imaging scans such as CT and the MNP localization imaging scan using MPI will be acquired and co-registered. MPI images can be used to calculate PID gains values more accurately from MPI data that can create spatially accurate images of magnetic nanoparticles in tissues, but these will require co-registration with anatomical imaging such as CT or MRI. Alternatively, MPI can be integrated with the MHT device to provide on-board imaging. In this case, MHT can be suspended to generated updated images that can be used as “online” input to refine treatment during each MHT session. In the online process, the patient would be scanned with onboard anatomical imaging, such as CBCT and PID gains would be updated accordingly.

A significant safety consideration that our PID controller design accommodated was off-target heating non-specific power deposited in tissue by Joule heating, i.e., heating resulting from circular eddy currents. This power depends on the tissue electrical conductivity (σ), AMF amplitude (*H*), frequency (*f*) and radius ^®^ of the eddy current path. The total power absorbed by tissue is proportional to *σ* × (*H* × *f* × *r*)^2^, with greatest power deposition occurring at the periphery. This non-uniform heating imposes safety limits on values of *H* and *f* for clinical MHT. Current consensus sets this limit to *H* × *f* < 4.85 × 10^8^ A/(m × s) for a 30 cm torso diameter [72]. In our current controller design, we limited non-specific heating by enabling user-defined upper limits on the temperature to override PID temperature controller function, reducing power when safety limits were exceeded.

Additional refinement of temperature control may be realized with power modulation (e.g., pulse width modulation) to harness physiological thermoregulatory responses that distribute heat more effectively and to enable more robust control of the temperature distribution within the tumor [26,27]. Though in vivo testing was conducted using only one live subject, multiple sessions were conducted. These were integrated with computational modeling and with in vitro and ex vivo (repeated) testing to provide confidence of the validation within FDA guidelines.

## 5. Conclusions

We verified and validated a feedback temperature controller, regulated by a proportional-integral-derivative (PID) algorithm, to control temperature during AMF-induced heating of MNPs. While calculated values served well as initial inputs, optimization of these was necessary during heating sessions for each system. This becomes critically important for MHT applications and 30-s “pre-heating conditioning” pulses were found to be invaluable to fine-tune the initial values of the input parameters. Future studies or development(s) that would enhance clinical adoption of MHT include integration of imaging of magnetic particles in tissue (e.g., magnetic particle imaging), volumetric thermometry, and adaptive controllers capable to incorporate multiple thermal inputs at locations throughout the tumor with adaptive controller algorithms.

## Figures and Tables

**Figure 1 cancers-15-00327-f001:**
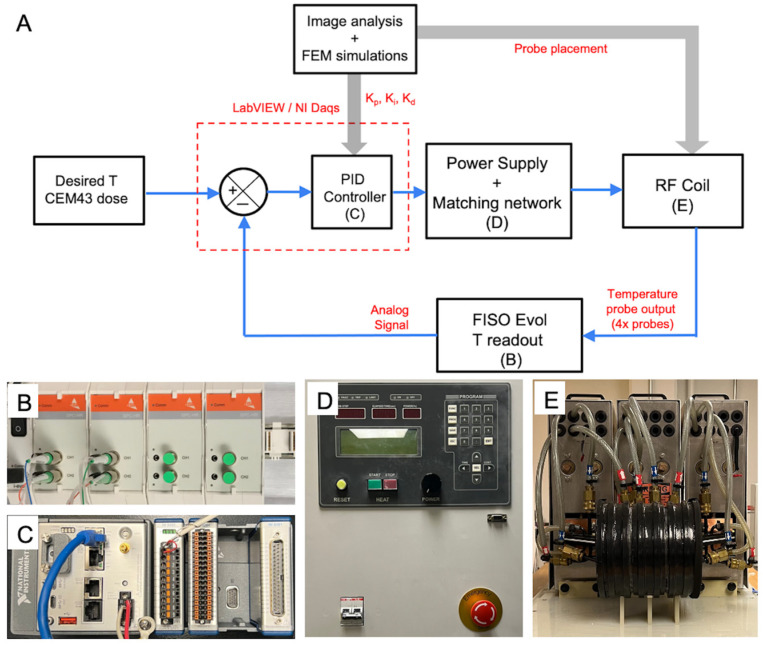
Experimental setup for temperature feedback control in magnetic hyperthermia (MHT) of canine glioblastoma. (**A**) Schematic block diagram depicting the core components, the flow of input, output signals and generation of control signals during feedback control of temperature during MHT. Core components include the (i) multi-sensor FISO EVO fiber optic probe for temperature measurement, (ii) FISO SPC-HR fiber optic reading module for conversion of temperature to analog outputs (**B**), (iii) a NI CompactRIO (NI cRIO 9042) controller with the multifunction I/O module and compatible LabVIEW software interface (**C**), (iv) 120 kW alternating magnetic field (AMF) induction heating power supply (**D**), and (v) a 20 cm diameter radiofrequency (RF) modified Maxwell coil applicator (**E**). User inputs to the LabVIEW interface include the setpoint temperature, $\;T_{ref}$, safety threshold temperature per measurement sensor (e.g., 50 °C to prevent ablation and runaway at treatment zone), second safety temperature threshold to prevent burns from non-specific Joule heating (e.g., 40 °C for probe placed at outer surface of brain), lower and upper voltage limits on the power supply for electrical safety, and proportional-integral-derivative (PID) controller gains.

**Figure 2 cancers-15-00327-f002:**
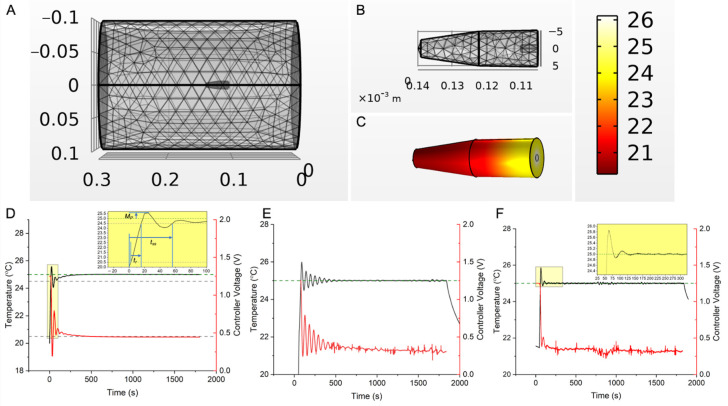
Temperature controller design, simulation, and experimental validation. (**A**) 3D mesh model of the RF coil and gel + Cu wire experimental setup, used to simulate open-loop and closed-loop temperature vs. time responses as a function of AMF and PID gains, through finite element time domain simulations of coupled electromagnetic and heat-transfer equations in COMSOL Multiphysics. (**B**) Zoomed in view of the 3D mesh model of the gel + Cu wire setup shown in (**A**). (**C**) Snapshot of the temperature distribution in the agarose gel + Cu wire system at 30 min, during a PID feedback-controlled simulation of heat transfer, where the setpoint temperature, $T_{ref}$, at the probe location (2.3 mm from center) was set to 25 °C. (**D**) Simulated temperature vs. time responses (black curve) for setup in (**C**), for input gains ($K_{p},\;\;K_{i\;},K_{d})$ of (*0.26*, *0.001*, *3*). Response curves to different input gains ($K_{p},\;\;K_{i\;},K_{d})$ in the neighborhood of calculated gain values (*0.26*, *0.001*, *0.11*) are shown in Appendix A (*0.26*, *0.001*, $K_{d}$ ), where 2 < $K_{d}\;$ < 3, was chosen as initial gains (starting with = 3) for the experimental system to ensure a quick rise time. The red curve and right-axis represent controller feedback voltage. A 0–1.25 V range correspond to 0–9.8 kA/m peak AMF amplitude at 160 kHz. Inset shows rise time (*t_r_*), settling time (*t_ss_*), and overshoot (*M_p_*). (**E**) Experimentally measured temperature vs. time response (black curve) in the agarose gel + Cu wire system, during a 30 min PID temperature-control experiment, where the setpoint temperature, $T_{ref}$, was input as 25 °C. Initial gains of (*0.2*, *0.001*, *3*), similar to the model, were entered, but integral gain$,K_{i}$, was lowered to lower the effect of integral-windup and consequent power supply tripping from rapid oscillations. Final gains that resulted in a stable compromise between rise time, settling time and minimizing power oscillations, were (*0.23*, *0.0001*, *2.84*). The red curve and right-axis represent controller feedback voltage. A 0–1.25 V range correspond to 0–9.8 kA/m peak AMF amplitude at 160 kHz. (**F**) Experimentally measured temperature vs. time response (black curve) in the “tuned” agarose gel + Cu wire system, during a 30 min PID temperature-control experiment, where the setpoint temperature, $T_{ref}$, was input as 25 °C and gains of (*0.23*, *0.0001*, *2.84*) were used. Inset shows the initial overshoot <5%. Settling time ($t_{ss}$ ) of 39.7 s and rise time ($t_{r}$ ) of 8.35 s were observed, well within the design specifications of the control system ($t_{ss}$ < 5 min, $t_{r}$ < 60 s). The red curve and right-axis represent controller feedback voltage. A 0–1.25 V range correspond to 0–9.8 kA/m peak AMF amplitude at 160 kHz. In vitro experiments for validation of controller temperature response were conducted three times (N = 3).

**Figure 3 cancers-15-00327-f003:**
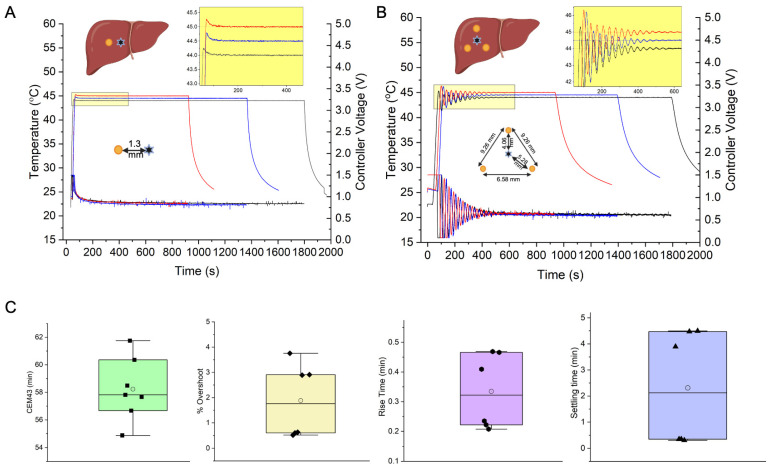
Temperature-controller performance during ex vivo heating experiments in a cow liver lobe, with single standard Cu wire heat source, (**A**), and a 3-wire distributed heat source configuration, (**B**). Yellow circle (●) in the liver schematic indicates a Cu wire cross-section, and a star symbol (✶) indicates fiberoptic temperature probe sensor placement. (**A**) Temperature vs. time (left *y*-axis), and controller voltage vs. time (right *y*-axis), responses of the PID controller to setpoint temperatures $T_{ref}$ of 44, 44.5 and 45 °C for treatment times of 30, 22.5 and 15 min, respectively in a cow liver lobe with a single Cu wire heat source. The probe sensor distance from the Cu heat source was 1.5 mm (±10%). Treatment times were chosen to cover a range of 15–30 min, based on clinical safety requirements (e/g. canine patient time under anesthesia <30 min) for MHT treatments. Based on the selected treatment time (15, 22.5 or 30 min), the setpoint temperatures were accordingly adjusted to achieve a target CEM43 of 60 min, i.e., 44 °C for 30 min, 44.5 °C for 22.5 min and 45 °C for 15 min. Three experiments were conducted for the single heat source geometry (N = 3). Within each experiment, a different target temperature setpoint was selected (44 °C for 30 min, 44.5 °C for 22.5 min and 45 °C for 15 min). Inset shows magnified view of the temperature overshoot (<1%) and steady state error (<1%). $\left(K_{p},\;\;K_{i\;},K_{d}\right)$ gains of (0.2, 1 × 10^−5^, 2.5) were used. (**B**) Temperature vs. time (left *y*-axis), and controller voltage vs. time (right *y*-axis), responses of the PID controller to setpoint temperatures $T_{ref}$ of 44, 44.5 and 45 °C for treatment times of 30, 22.5 and 15 min, respectively in a cow liver lobe with three distributed Cu wire heat sources. The probe sensor is placed within the triangle formed from the Cu wire cross-sections. Distances between Cu wires and between Cu wire and temperature sensor are given in the inset. Treatment times were chosen to cover a range of 15–30 min, based on clinical safety requirements (e.g., canine patient time under anesthesia <30 min) for MHT treatments. Based on the selected treatment time (15, 22.5 or 30 min), the setpoint temperatures were accordingly adjusted to achieve a target CEM43 of 60 min, i.e., 44 °C for 30 min, 44.5 °C for 22.5 min and 45 °C for15 min. Three experiments were conducted for the distributed heat source geometry (N = 3). Within each experiment, a different target temperature setpoint was selected (44 °C for 30 min, 44.5 °C for 22.5 min and 45 °C for 15 min). Inset shows magnified view of the temperature overshoot (<5%) and steady state error (<1%). ($K_{p}\;,K_{i\;},K_{d})$ gains of (0.5, 1 × 10^−6^, 2.5) were used. (**C**) Box-whisker + scatter plot of controller performance parameters obtained from temperature vs. time data in (**A**,**B**). Performance metrics include efficacy in attaining target treatment specifications of cumulative equivalent minutes at 43 °C (CEM43) reported in min, and design specifications of settling time ($t_{ss})$, rise time ($t_{r}$ ) and percentage overshoot ($\%M_{p}$ ). Lower settling times (<1 min) and overshoot (<1%) were observed in the single heat sources case vs. multiple heat sources. (**C**) Box whisker plots of CEM43 values calculated from measured temperatures.

**Figure 4 cancers-15-00327-f004:**
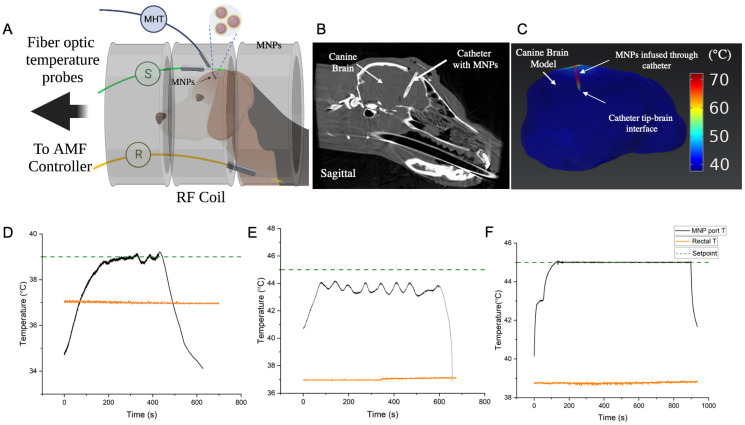
(**A**) Schematic depicting setup for in vivo validation of MHT controller in a canine research subject. Magnetic nanoparticles (MNPs) (cFe ~80 mg Fe/mL, Synomag, Micromod Partikeltechnologie, GmbH) were infused into the canine brain through a custom catheter. Fiberoptic temperature probes were inserted at specific locations for feedback control and safety. One temperature probe was inserted through the catheter’s MNP port into the brain tissue (MHT, black), with the temperature sensor localized closest to the MNP heat source. This sensor was used for feedback temperature control at that location. A second temperature probe was placed in the rectum to monitor core body temperature (R, orange) and a third probe was inserted subcutaneously in the canine head (S, green) to monitor temperature increase from eddy current heating during the treatment. The temperature setpoint for the MHT probe, and safety limits for the rectal and subcutaneous probes were input into the controller program before the treatment. (**B**) Cone-beam computed tomography (CT) image of the canine brain showing the catheter with infused MNPs localized within and around the catheter. (**C**) Finite element based coupled electromagnetic and heat transfer simulations performed on the CT image-segmented brain and MNP heat sources (within and surrounding the catheter walls) showed that at 20 kA/m peak, 160 kHz, heating occurred mostly at the catheter where the MNPs were located. Temperature increase, ΔT, away from the MNP heat sources (>3 mm from the catheter walls) was negligible due to the heat sink effects of perfusion. An injected MNP concentration of cFe = 80 mg Fe/mL was assumed, with a normal distribution about the catheter axis with standard deviation σ=3 mm. (**D**) Temperature vs. time response at the brain tissue adjoining the MNP catheter tip, for a setpoint of 39 °C, showed that setpoint was achieved within 200 s, however small temperature fluctuations (<0.4 °C) were observed. Rectal temperature remained stable at ~37 °C throughout this test. This test was performed for preliminary safety evaluation of exposing the dog to AMF. (**E**) Temperature vs. time response at the brain tissue adjoining the catheter ‘s MNP port-tip, for a hyperthermic setpoint of 45 °C, showed that the setpoint was not achieved in this 10 min test. The temperature at this location increased to ~44 °C and fluctuated between 43–44 °C. The fluctuating response, typical of an untuned controller, was attributed to non-optimal PID gains. Core body temperature (rectal) was stable at ~37 °C. (**F**) Temperature vs. time response at the brain tissue adjoining the catheter’s MNP port-tip, after optimal tuning of the controller PID gains, showed that the controller achieved the setpoint of 45 degrees C in ~120 s and remained stable (Δ*T* < 0.1 °C) throughout the remainder of the 15 min treatment. Final PID gains of (2, 1 × 10^−4^, 2.5) were used. CEM43 isoeffect thermal dose for the treatment was 56 min. Rectal temperature remined stable at ~38.5–39 °C during the entire treatment. Temperature increases at MHT sensor location (near MNP) to pulses of increasing AMF amplitude were used for fine tuning of PID gains prior to the 15 min treatment and are shown in Appendix A. Thus, a total of three short duration AMF pulse tests and three longer temperature-control tests (N = 3, (**D**–**F**)) were performed. Temperature vs. time plots, corresponding to eddy current heating, measured at the subcutaneous location on the canine head (green probe in (**A**)) for cases 4 (**B**–**D**) are shown in Appendix A.

## Data Availability

Data are available upon request.

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
