# Peer review of "Validation of a Temperature-Feedback Controlled Automated Magnetic Hyperthermia Therapy Device"

_cancers, 2023, doi:10.3390/cancers15020327_

Round 1

Reviewer 1 Report

The paper presents a study of validation of an instrumentation that temperature-feedback controlled automated magnetic hyperthermia therapy device. The authors development an algorithm based on proportional-integral-derivate (PID) protocol for controlled of temperature in MHT device. The validation were performed in experiments ex vivo (bovine liver lobe) and in vivo, in a healthy, adult male beagle (~10 kg) with a custom catheter for MNP delivery in left frontal lobe of the brain was studied.

The results show that a feedback temperature controller, regulated by a (PID) algorithm, exhibit good results for to control temperature during AMF-induced heating of MNPs.

In my point of view, the paper is suitable for published in Cancers

Author Response

REVIEWER 1

Comments and Suggestions for Authors

The paper presents a study of validation of an instrumentation that temperature-feedback controlled automated magnetic hyperthermia therapy device. The authors development an algorithm based on proportional-integral-derivate (PID) protocol for controlled of temperature in MHT device. The validation were performed in experiments ex vivo (bovine liver lobe) and in vivo, in a healthy, adult male beagle (~10 kg) with a custom catheter for MNP delivery in left frontal lobe of the brain was studied.

The results show that a feedback temperature controller, regulated by a (PID) algorithm, exhibit good results for to control temperature during AMF-induced heating of MNPs.

In my point of view, the paper is suitable for published in Cancers

Response: We thank the reviewer for the compliments and comments.

Reviewer 2 Report

Dear Authors;

Re: Manuscript ID: cancers-2060303

"Validation of a temperature-feedback controlled automated magnetic hyperthermia therapy device"

I enjoyed reading your manuscript which aimed to design and validate a PID-based temperature feedback control system in MHT. Please find my comments and suggestions below:

1. In Abstract please describe abbreviations, e.g., MNPs, CEM.

2. Please check the sentence in Lines 52-53 (... high LET RT) and rephrase.

3. Number of trials / tests need to be clearly mentioned in relevant sections and presented data.

4. Figure legends are descriptive, however some abbreviations and left undescribed (e.g., in Figure 4 MHT, MNP, ...).

5. References from current year are very limited (I was only able to find 2). Please add more studies from 2022.

6. In the discussion of prior art pertaining to simulation of MHT consult recent papers (2023 - Ahead of Press) such as: 

https://doi.org/10.1016/j.materresbull.2022.112035

7. A couple of lines at the end of conclusions containing your suggestions for future studies will highly benefit the readers.

8. Reference number 35 requires "Date Visited".

Thank you

Author Response

REVIEWER 2

Comments and Suggestions for Authors

Dear Authors;

Re: Manuscript ID: cancers-2060303

"Validation of a temperature-feedback controlled automated magnetic hyperthermia therapy device"

I enjoyed reading your manuscript which aimed to design and validate a PID-based temperature feedback control system in MHT. Please find my comments and suggestions below:

Comment 1: In Abstract please describe abbreviations, e.g., MNPs, CEM.

Response: We thank the reviewer for the helpful comment. We have revised the abstract as suggested.

Comment 2:  Please check the sentence in Lines 52-53 (... high LET RT) and rephrase.

Response: We thank the reviewer for pointing out our lack of clarity. We have revised the manuscript text accordingly to enhance clarity for readers.

Comment 3:  Number of trials / tests need to be clearly mentioned in relevant sections and presented data.

Response: We thank the reviewer for pointing out our oversight. We have revised the Methods section in the text to inform readers of numbers of replicate experiments.

Comment 4:  Figure legends are descriptive, however some abbreviations and left undescribed (e.g., in Figure 4 MHT, MNP, ...).

Response: We thank the reviewer for the suggestion. The first appearance of an acronym in figures has been defined.

Comment 5:  References from current year are very limited (I was only able to find 2). Please add more studies from 2022.

Response: We thank the reviewer for pointing out our oversight. We have included a few selected additional references that have particular relevance to the current work, as suggested by the reviewer. We note that our search produced 745 published works in the period 2021-2023 in Clarivate Web of Science (search terms – “magnetic hyperthermia” AND “heating”). As our manuscript is not a review or meta-analysis paper, we have included only a few of these (review and original works) that have particular relevance. We acknowledge some may have been excluded but this is unavoidable to maintain reader focus.

Comment 6:  In the discussion of prior art pertaining to simulation of MHT consult recent papers (2023 - Ahead of Press) such as: 

https://doi.org/10.1016/j.materresbull.2022.112035

Response: We thank the reviewer for the helpful suggestion. We have included additional references that have particular relevance to the current work, as suggested by the reviewer. However, we wish to point out that our search produced over 166 published works in the period 2021-2023 in Clarivate Web of Science (search terms – “magnetic hyperthermia” AND “modelling” AND “heating”). As our manuscript is an original research publication, we have included a few, and included the one suggested by the reviewer.

Comment 7:  A couple of lines at the end of conclusions containing your suggestions for future studies will highly benefit the readers.

Response: We thank the reviewer for the helpful suggestion. We have added a sentence for benefit of readers as suggested.

Comment 8:  Reference number 35 requires "Date Visited".

Response: Done.

We thank the reviewer for the comments and identifying our oversights to help us improve the reader’s experience. 
